# Apoptotic Effect of Combinations of T-2, HT-2, and Diacetoxyscirpenol on Human Jurkat T Cells

**DOI:** 10.3390/toxins17040203

**Published:** 2025-04-18

**Authors:** Phattarawadee Wattanasuntorn, Saranya Poapolathep, Patchara Phuektes, Imourana Alassane-Kpembi, Johanna Fink-Gremmels, Isabelle P. Oswald, Amnart Poapolathep

**Affiliations:** 1Interdisciplinary Graduate Program in Genetic Engineering, Graduate School, Kasetsart University, Bangkok 10900, Thailand; phattarawadee.oil2557@gmail.com; 2Department of Pharmacology, Faculty of Veterinary Medicine, Kasetsart University, Bangkok 10900, Thailand; fvetsys@ku.ac.th; 3Department of Pathobiology, Faculty of Veterinary Medicine, Khon Kaen University, Khon Kaen 40002, Thailand; patphu@kku.ac.th; 4Department of Veterinary Biomedicine, Faculty of Veterinary Medicine, Université de Montréal, Saint-Hyacinthe, QC J2R 0A8, Canada; imourana.alassane-kpembi@umontreal.ca; 5Institute for Risk Assessment Sciences, Faculty of Veterinary Medicine, Utrecht University, 3508 Utrecht, The Netherlands; j.fink@uu.nl; 6Toxalim (Research Centre in Food Toxicology), Toulouse University, INRAE, ENVT, INP-Purpon, UPS, 31000 Toulouse, France; isabelle.oswald@inrae.fr

**Keywords:** trichothecene type A, combination index, human Jurkat T cell, cytotoxicity

## Abstract

Trichothecene type A mycotoxins, such as T-2, HT-2, and diacetoxyscirpenol (DAS), are known to induce cytotoxicity and apoptosis in different cell types. As all three Fusarium toxins may occur concomitantly in a given food or feed commodity, there is growing interest in the effect of such mycotoxin mixtures. This study aimed to identify the toxic interactions among T-2, HT-2, and DAS in a human Jurkat cell model. As a first step, an MTT assay was used to assess cytotoxicity after 24 h of cell exposure to individual mycotoxins and their mixtures. The results were used to calculate the combination index (CI), which indicates the nature of the mycotoxin interactions. In Jurkat T cells, the toxicity ranking for the individual mycotoxins was T-2 > HT-2 > DAS. The CI values of the dual and triple mycotoxin combinations calculated from the results of the MTT and reactive oxygen species assays showed synergistic effects at low concentrations and an apparent antagonism at very high concentrations for all combinations. The additional cytometric analyses confirmed the synergistic effects, as expected, following co-exposure to the three tested trichothecenes. As the lower toxin concentrations investigated reflect natural contamination levels in food and feeds, the synergistic effects identified should be considered in risk characterization for trichothecene exposure in humans and animals.

## 1. Introduction

Mycotoxins are currently considered as the most prevalent undesirable contaminants of animal feeds and human food supplies [1,2]. Of particular interest are mycotoxins produced by Fusarium species due to their worldwide occurrence in all climatic zones [3].

The major class of Fusarium toxins is trichothecenes (TCTs), a group consisting of about 180 individual toxins, which are commonly divided according to their chemical structure into types A, B, C, and D TCTs [4]. Representative of TCT-A are T-2 toxin, HT-2 toxin, and diacetoxyscirpenol (DAS) often produced simultaneously by *F. sporotrichioides* [5,6], which is frequently found in cereals and animal feeds. Recent studies indicate that the contamination levels of T-2 and HT-2 toxins in foods and feeds vary significantly across different regions and commodities. In Northern Europe, particularly in the United Kingdom, oats have exhibited high concentrations of these toxins, with mean levels reaching up to 570 µg/kg and maximum levels as high as 9990 µg/kg. Similarly, in Scotland, a 2019 survey found T-2 and HT-2 toxins in 91% of oat samples, with concentrations ranging from being non-detectable to 3474 µg/kg. In contrast, data from Thailand between 2015 and 2020 show that T-2 toxin contamination in animal feeds was predominantly low, with 94.7% of samples containing less than 25 µg/kg and no samples exceeding 250 µg/kg. Information on diacetoxyscirpenol (DAS) contamination is less prevalent, but it is generally detected at lower frequencies and concentrations compared to T-2 and HT-2 toxins [6].

TCTs have been shown to be potent inhibitors of protein synthesis and can affect mitochondrial function, and they are inhibitors of nucleotide synthesis. In vivo, TCTs induce immunosuppression and oxidative stress [7]. For example, T-2 toxin is considered the most potent toxic TCT; its effects on lymphocytes are well documented, with [8] and [9] demonstrating that T-2 toxin induces apoptosis in T lymphocytes through mitochondrial damage and caspase activation. A pro-apoptotic effect has also been shown in Jurkat cells. Another study presented in [10] further elucidated that T-2 toxin exacerbates oxidative stress, which plays a key role in cytotoxicity. Several studies have demonstrated that HT-2 binds to ribosomes, leading to a ribotoxic stress, inhibition of protein synthesis, cell cycle arrest, and decreased cell proliferation [11,12,13]. In Jurkat models, HT-2 toxin impairs T cell activation and cytokine production, as reported in [14]. DAS, another TCT-A, has been less studied compared to T-2 and HT-2 toxins. The authors of [15] reported that DAS causes significant damage to lymphocytes by inducing apoptosis and disrupting protein synthesis. DAS exerts its toxic effects by enhancing oxidative stress and disrupting cellular metabolism, as highlighted in [16].

The biosynthesis of TCT by Fusarium is regulated by a defined gene cluster, which can result in the formation of a different type of TCT-A [3,17]. Hence, it is highly likely that a single grain sample may contain different mycotoxins, making it mandatory to include the potential interaction of such closely related mycotoxins into the overall risk assessment related to human and animal exposure in real life. Several investigations have already indicated that simultaneous exposure to multiple TCTs can lead to synergistic toxic effects, such as exacerbating inflammation, as well as oxidative stress and increasing cellular damage beyond the sum of the effects of individual toxins [18,19,20,21,22,23].

As mentioned above, the immune system is one of the primary targets of TCT-A, and their effects on immune cells, particularly T lymphocytes, have been studied in different experimental models. The Jurkat cell, a human T lymphocyte cell line, serves as an established in vitro model to study the immunotoxicity and cytotoxic mechanisms of environmental and dietary toxins [24]. Previous studies have demonstrated that exposure to an individual TCT can lead to increased production of ROS, mitochondrial dysfunction, and activation of apoptotic pathways in Jurkat cells [25,26,27].

The aim of the current experiment was to study the effect of mixtures of the three main TCT-A mycotoxins. The most relevant aspect in the study of mixtures is the analysis of the dose response to identify synergistic or antagonistic effects. If the tested mycotoxins do not interact with each other, their effects can be described as additive, while non-linear relations may result in synergistic or antagonistic effects.

For this characterization of combined effects, a range of assays were used to measure cell viability, cell morphology, and apoptosis.

## 2. Results

### 2.1. Comparative Cytotoxicity of T-2, HT-2, and DAS on Human Jurkat T Cells

To assess the cytotoxic effects of T-2, HT-2, and DAS—both individually and in combination—a colorimetric MTT assay was used, which measures mitochondrial succinate dehydrogenase activity as an indicator of cell viability. Jurkat T cells were exposed to different concentrations of the toxins for 24 h. The results show a dose-dependent reduction in cell viability. Among the tested mycotoxins, T-2 exhibited the highest cytotoxicity, followed by HT-2 and then DAS (Figure 1).

### 2.2. Combined Cytotoxic Effects of T-2, HT-2, and DAS on Human Jurkat T Cells

The dose–effect relationship curves for the toxicity of the mycotoxins alone and in mixtures are presented in Figure 1 and Figure 2. The results show that cell viability decreased significantly in a dose-dependent manner when exposed to single toxins and their mixtures. At all tested concentrations, the viability counts of cells exposed to the T-2 + HT-2, T-2 + DAS, HT-2 + DAS, and T-2 + HT-2 + DAS mixtures were lower than those for the dose responses of individual toxins.

Table 1 lists the parameters of the dose–effect relationship derived from in vitro viability studies. The correlation coefficients (r) were obtained from median–effect plots. The Dm values obtained in the mixtures showed the strongest effect for T-2 + HT-2 + DAS (0.0026 µg/mL; 0.063 µM). In the dual combinations, T-2 + HT-2 (0.3266 µg/mL; 0.733 µM), T-2 + DAS (0.5859 µg/mL; 1.38 µM), and HT-2 + DAS (39.5639 µg/mL; 103.08 µM) produced the weakest effects.

### 2.3. Analysis of Interaction of T-2, HT-2, and DAS When Applied as Mixtures

As the interactions between multiple mycotoxins can be synergistic, additive, and antagonistic, the CI values were calculated. The CI/fa curves, along with the 95% CI, are displayed in Figure 3 and Table 2 for the various mycotoxin combinations that were examined. The dual mixtures of T-2 + DAS and HT-2 + DAS, as well as the triple combinations of T-2 + HT-2 + DAS, had similar patterns as a function of mycotoxin concentration. Low doses produced synergistic effects and medium doses were additive, while antagonistic interactions were observed at high mycotoxin concentrations. In contrast, the dual mixture of T-2 + HT-2 displayed a different interaction pattern, where low doses showed antagonistic effects, medium doses were synergistic, and high doses were antagonistic.

To further quantify the synergistic interaction between T-2, HT-2, and DAS, the DRI values were evaluated. The DRI values for the synergistic effects of the T-2 + HT-2, T-2 + DAS, and HT-2 + DAS mixtures varied in the range of 0.00005–923 at low toxicity levels (25% toxicity). In the triple combination, the DRI value for the synergistic effect varied from 0.00003 at low toxicity levels to 2172 at medium toxicity levels. The DRI indicated the dose reduction multiplier of the combined dose of the tested toxins compared to the dose of each toxin at the same level of inhibition. The DRI values for the HT-2 + DAS and T-2 + HT-2 + DAS mixtures at a 10% toxicity level were numerically the opposite to the CI values (Table 2). When evaluating the enhanced cytotoxicity of the three mycotoxins, CI < 1 suggests a synergistic effect, indicating that the combination of toxins enhance cytotoxicity beyond what would be expected by the simple addition of their individual single effects. However, DRI < 1 indicates that a combination of toxins is less potent than the sum of individual toxicity levels, indicating apparent antagonisms.

### 2.4. Effects of T-2, HT-2, and DAS and Their Combinations on Cellular Oxidative Stress

Oxidative stress is triggered by an increase in the intracellular levels of reactive oxygen species (ROS), damaging lipids, proteins, and DNA, and hence, it is a common cause of cell death. Based on the results of the current study, toxin mixtures caused higher levels of ROS than the individual toxins. The triple combination induced the highest ROS level, significantly exceeding those observed with any single toxin or dual combinations (Figure 4).

### 2.5. Determination of Apoptotic Cells Using Fluorescence Microscopy Exposed to Combined Mycotoxins

The extent of apoptosis induced by co-exposure to two or three mycotoxins was evaluated using fluorescence microscopy with Apopxin™ Green staining. Based on the results (Figure 5), the combination of T-2 + HT-2 or of T-2 + DAS induced significant increases in apoptotic cells compared to the untreated controls. Co-exposure to T-2 and DAS also led to a substantial increase in apoptosis, though it was slightly less pronounced than with the T-2 + HT-2 and T-2 +DAS combinations. The triple combination produced the highest apoptotic response (Figure 5).

### 2.6. Detection of Apoptotic Rate Using Flow Cytometer of Human Jurkat T Cells Exposed to Combinations of T-2, HT-2, and DAS

The flow cytometry analysis using Annexin V/PI staining revealed a significant increase in apoptosis in human Jurkat T cells after co-exposure. Compared to the control group (which displayed a minimal apoptotic rate), the treated groups had a dose-dependent elevation in both their early and late apoptotic populations. The dual combinations (T-2 + HT-2, T-2 + DAS, and HT-2 + DAS) produced synergistic effects. Among the dual combinations, T-2 + HT-2 produced the highest apoptotic rate (Figure 6A,B).

The triple combination of T-2, HT-2, and DAS induced the most significant apoptotic response (Figure 6B). This effect was evident in the elevated early apoptotic cell population, indicating rapid initiation of the apoptotic pathway. The increase in late apoptotic and necrotic cells further supported the severe cytotoxicity caused by combined exposure.

## 3. Discussion

The objective of the current study was to re-assess and characterize the cytotoxic and apoptotic effects of three prominent Fusarium toxins, including T-2, HT-2, and DAS. The cytotoxic effects of these TCT-A toxins are well documented, affecting several cell functions, such as cell proliferation and viability, for example, in male and female reproductive cells, hematopoietic stem cells, and immune cells [28,29]. The current study investigated the effects of these toxins alone or in dual or triple combination on human Jurkat T cells. As a first step, cell viability was assessed using the MTT test. The MTT test is linked to mitochondrial function and was applied to quantify cell viability. The results show that all three mycotoxins exerted an immediate cytotoxic effect, albeit with slight differences in their potency.

Based on the IC_50_ values for each mycotoxin, the T-2 toxin was identified as the most toxic to human Jurkat T cells. The effective concentration of the T-2 toxin required to inhibit cell viability by 50% was consistent with values reported in other cytotoxicity studies. For example, in human lymphocytes, the T-2 toxin reduced mitogen-induced proliferation, with IC_50_ values in the range of 0.9–1.3 nM [30]. Similarly, porcine alveolar macrophages exposed to T-2 toxin for 16 hr had an IC_50_ value of approximately 10 nM [28]. In other cell types, HT-2 exerted significantly lower cytotoxic effects on human renal proximal tubule epithelial cells, normal human lung fibroblasts, and mouse fibroblasts [31,32]. Variations in cell sensitivity were attributed to various experimental factors, such as cell lines, culture medium composition, supplements, exposure duration, and toxin concentration ranges. The cytotoxic concentration for DAS in human Jurkat T cells ranged between 0.01 and 0.15 μM, which is also exerted by apoptotic cell death as well as cell cycle arrest in human Jurkat T cells [33].

For the evaluation of the combined effect of three toxins, following the co-exposure of human Jurkat T cells, the Chou–Talalay method was used, which was originally developed to assess drug–drug interactions. Dual combinations (T-2 + HT-2, T-2 + DAS, or HT-2 + DAS) and the triple combination (T-2 + HT-2 + DAS) produced synergistic effects at low cytotoxicity levels (25–50% inhibition) and antagonistic effects at higher cytotoxicity levels (>50% inhibition); however, a DRI < 1 indicated that higher concentrations of each toxin were required in the combination to achieve the same effect as the sum of individual toxins. This indicated that at low concentrations tested, which are representative of feed and food contamination levels, a synergistic effect needs to be expected following co-contamination. At higher mycotoxin concentrations, also used in this in vitro assay, apparent antagonisms were observed when applying the mass action law-based algorithm of Chou [34]. Similar synergistic interactions were observed in other studies [19,35]. For example, CI values in the range of 0.4–0.7 were reported from experiments with CaCo-2 cells exposed to T-2 and HT-2, with evidence that in addition to mitochondrial dysfunction and apoptosis, these cells produced significant disruption of tight junction integrity, highlighting tissue-specific vulnerabilities [36,37]. Primary hepatocytes, known for their high metabolic activity, are also highly susceptible to TCT-A [38]. A strong synergistic effect (CI < 0.5) between T-2 and HT-2 was observed with combined exposure, exacerbating oxidative stress and lipid peroxidation. Collectively, these results emphasize that co-exposure to multiple TCT-A mycotoxins is likely to exert synergistic effects, which may vary depending on the cell type, toxin combination, and concentration.

Apoptosis begins when cells are damaged to an extent that physiological repair is unlikely or not possible; the process advances through distinct stages, usually involving cell rounding, shrinkage, chromatin condensation, nuclear fragmentation, and membrane blebbing, ultimately resulting in cell death [39,40,41,42]. In this research, the morphological changes in the cells exposed to single toxins were compared with the dual or triple combinations of the selected toxins. Fluorescence microscopy effectively highlights both the individual and synergistic effects of TCT-A on cellular morphology. Mycotoxin exposed Jurkat T cells showed typical apoptotic features, such as chromatin condensation, characterized by increased fluorescence intensity due to DNA fragmentation. Membrane blebbing and cytoplasmic shrinkage were also evident, particularly at higher toxin concentrations. These changes align with the known effects of TCT-A mycotoxins, disrupting protein synthesis and inducing oxidative stress, which ultimately lead to apoptosis via mitochondrial dysfunction [42].

In the current study, the combined treatments with two or three mycotoxins led to a higher percentage of apoptotic cells than in the individual toxin treatments, suggesting synergism. This observation is consistent with observations in other cell types, such as CaCo-2 cells [43]. The results of the current study further suggest that the synergistic effects of combined exposure to T-2, HT-2, and DAS may be directly related to the induction of oxidative stress, amplifying cellular stress and inducing apoptosis. Therefore, ROS measurements were performed in this study for all individual toxins and toxin combinations, which were later used in the assessment of apoptosis. Co-exposure to multiple TCT-A mycotoxins produced synergistic effects in increasing ROS levels so that they were significantly higher than those observed with individual toxins. These findings align with similar synergistic oxidative stress responses in lymphocytes exposed to T-2 and HT-2 [44]. High levels of ROS are observed when the physiological cellular defense mechanisms are depleted, and hence, cell injury is beyond physiological repair mechanisms and cells undergo programmed cell death [45,46].

Various studies have provided additional evidence that ROS formation is one of the most important features of TCTs, particularly TCT-A. Therefore, ROS formation, and the inhibition of key antioxidant enzymes, including superoxide dismutase (SOD) and catalase, were reported by several authors [47,48,49,50,51,52,53] and associated with mitochondrial dysfunction, for example, for T-2 [48,49]. In this context, it was also shown that TCT-A disrupts mitochondrial membrane potential and activates NADPH oxidases, leading to amplified ROS production [49]. Also, in human Jurkat T cells, mitochondrial dysfunction in ROS dynamics and impaired oxidative phosphorylation were reported following exposure to TCT-A [50]. The synergistic effects of T-2 and HT-2 on ROS production and apoptosis were also previously observed, and there was enhanced oxidative stress in cells co-exposed following co-exposure [51].

The current study used flow cytometry to investigate the combined cytotoxic effects of TCT-A on the apoptotic rate in human Jurkat T cells. Flow cytometry, known for its high sensitivity and specificity in analyzing apoptosis, enabled the precise evaluation of early and late apoptotic events. Based on this analysis, the current results show a clear dose-dependent increase in apoptosis rates after exposure to individual toxins. In agreement with other studies, synergistic apoptotic effects could be demonstrated when human Jurkat T cells were exposed to toxin combinations. A marked increase in Annexin V-positive/PI-negative cells (early apoptosis) and Annexin V-positive/PI-positive cells (late apoptosis) highlighted the progression of apoptosis in the current experiments. In consideration of the crucial role of ROS formation, the administration of antioxidants is likely to reverse many of the adverse effects exerted by TCT-A [52,53].

In summary, the current study demonstrated synergism in the severity of toxic effects when human Jurkat T cells were exposed to TCT-A family toxins, such as T-2 toxin and HT-2 toxin, which were selected for these experiments. Synergism was consistently demonstrated for all parameters tested, including cell viability, ROS formation, and the induction of apoptosis.

## 4. Conclusions

The combined exposure to TCT-A mycotoxins (T-2, HT-2, and DAS) significantly enhanced the cytotoxic effects in human Jurkat T cells compared to individual toxin treatments. The observed effects included increased ROS generation and apoptosis, indicating that these mycotoxins act synergistically to disrupt cellular homeostasis. These findings emphasize the need to include realistic exposure scenarios, such as co-exposure to multiple mycotoxins, into a comprehensive risk assessment and in the characterization of the remaining uncertainties when setting maximum limits. The synergistic interactions among these toxins pose an additional hazard to human and animal health, underscoring the importance of using multi-toxin methods for the monitoring of the rate of contamination of food and feed commodities.

## 5. Materials and Methods

### 5.1. Chemicals and Reagents

Standards for T-2 (C_24_H_34_O_39_, 466.52 g/mol, 98% purity), HT-2 (C_22_H_32_O_8_, 424.48 g/mol, 98% purity), and DAS (C_19_H_26_O_7_, 366.41 g/mol, 98% purity) were acquired from Sigma-Aldrich Merck (Sigma-Aldrich Merck, Darmstadt, Germany). These were dissolved in dimethyl sulfoxide (DMSO) to prepare 1 mM stock solutions and stored at −20 °C in the dark. The 3-(4,5-dimethylthiazol-2-yl)-2,5-diphenyltetrazolium bromide (MTT) and Annexin V-FITC apoptosis detection kits were also purchased from Sigma-Aldrich Merck (Sigma-Aldrich Merck, Darmstadt, Germany). The Total ROS Assay Kit (520 nm) was sourced from ThermoFisher Scientific (ThermoFisher Scientific, Waltham, MA, USA). RPMI-1640 medium, fetal bovine serum (FBS), and penicillin/streptomycin were purchased from Gibco Invitrogen (Gibco Invitrogen, Breda, The Netherlands). The Cell Meter™ Apoptotic and Necrotic Multiplexing Detection Kit I (Ternary Fluorescence Colors) was acquired from AAT Bioquest (AAT Bioquest, Pleasanton, CA, USA).

### 5.2. Cell Culture and Treatments

T cells (catalog number: EP-CL-0129; Elabscience; Houston, TX, USA) were maintained in RPMI-1640 medium containing 10% fetal bovine serum and 1% penicillin/streptomycin. The cells were grown at 37 °C in a 5% CO_2_ humidified environment, with the medium being changed every two days. Cells were seeded in 96-well plates at a density of 1 × 10^5^ cells/well and exposed to various concentrations of individual and combined mycotoxins. The concentrations of mycotoxins for individual exposure were in the ranges 0.11–5.36 µM for T-2, 0.12–5.89 µM for HT-2, and 0.14–6.82 µM for DAS.

The ratios of each mycotoxin in the dual and triple combinations were based on their individual 50% inhibitory concentration (IC_50_) values [27] from the previous study, which resulted in approximately equipotent toxicity for each mycotoxin in the mixture. In the control group, the cells were treated with 0.4% DMSO culture medium, while the blank group consisted of culture medium without any cells.

### 5.3. Evaluation of Cytotoxic Effects

Improved MTT^®^ assays sourced from Sigma-Aldrich MERCK (Darmstadt, Germany) were used to study cell viability in the presence of the single mycotoxins and their mixtures. Briefly, human Jurkat T cells were seeded in 96-well plates at a density of 1 × 10^5^ cells/well. After 24 h, when the cells had reached 70–80% confluence, the culture medium was replaced with fresh medium. After that, the cells were exposed to different concentrations of T-2, HT-2, and DAS and incubated for 24 h. Subsequently, 10 µL of MTT solution was added into each well, and the cells were incubated at 37 °C for 4 h. A blank control group (no cells) and three replicates for each individual treatment were assayed for each group. The optical density was measured using a microplate reader (Multiskan GO; ThermoFisher; MO, USA) at a wavelength of 450 nm. The percentage of viable cells was calculated using the following formula: $$Cell\;viability\left(\%\right)=100\times \frac{OD\;of\;treaded\;samples-Average\;OD\;of\;blanks}{OD\;of\;Control-Average\;OD\;of\;blanks}$$
where OD is the optical density at 450 nm, and the treated sample is either one of the tested mycotoxins or one of their tested combinations [27].

### 5.4. Identification of Apoptotic Cells Through Fluorescence Microscopy and Flow Cytometric Analysis

Apoptotic cell levels were determined using Apopxin™ Green and examined under a fluorescence microscope equipped with an FITC (fluorescein) filter. Human Jurkat T cells (1 × 10^6^ cells/well) were seeded into 24-well plates and incubated at 37 °C with increasing concentrations of T-2, HT-2, and DAS or their dual and triple combinations for 24 h. After each treatment, cells were rinsed using phosphate-buffered saline, labeled with 100× Apopxin™ Green assay solution, and left to incubate at room temperature for 30 to 60 min. Selective staining differentiated apoptotic, viable, and non-viable cells, and they were assessed. A Texas Red filter was employed for detecting 7-Aminoactinomycin D (7-AAD) and/or DAPI staining, whereas CytoCalcein™ Violet 450 staining was visualized using a violet filter. Fluorescent images were captured using a Zeiss AXIO microscope (AXIO; Zeiss; Oberkochen, Germany). Apoptotic cells appeared green, live cells appeared blue, and necrotic cells appeared red. Cell counts were determined based on these colors.

Jurkat cells were plated at a density of 1 × 10^5^ cells per milliliter in 6-well culture plates. The cells were subjected to multiple concentrations of T-2, HT-2, or DAS and their dual and triple combinations for 24 h. The apoptotic cells were measured based on Annexin V-FITC/PI co-staining assay. After each treatment, the cells were harvested and centrifuged at 1600 rpm for 10 min. Then, the produced pellet was re-suspended in 50 µL binding buffer containing 5 µL of Anexin V-FITC and incubated at 4 °C for 30 min in the dark. Propidium iodide (PI) in 200 µL of binding buffer was added to each of the tubes and incubated at room temperature for 10 min. Finally, the cells were analyzed using a BD FACSCanto™ II Flow Cytometer (Becton Dickinson; Franklin Lakes, NJ, USA), recording 30,000 events per sample [27].

### 5.5. Experimental Design and Assessment of Effect of Mycotoxin Combinations

#### 5.5.1. Experimental Design

Solutions of T-2, HT-2, and DAS (prepared as described above) were used for the individual, dual (T-2:HT-2; T-2:DAS; HT-2:DAS), and triple (T-2:HT-2:DAS) combinations. Human Jurkat T cells were exposed with serial dilutions of each mycotoxin individually and with a fixed constant ratio (T-2:HT-2 ratio = 0.1:0.5 and 0.5:1; T-2:DAS ratio = 0.1:0.5 and 0.5:2.5; HT-2:DAS ratio = 0.25:0.5 and 1:2.5; T-2:HT-2: DAS ratio = 0.1:0.25:0.5 and 0.5:1:2.5) based on the individual IC_50_ values in their dual and triple combinations. The control consisted of cells incubated with just the cell culture medium, while the blank group contained only the medium without cells.

#### 5.5.2. Assessment of Cytotoxic Effects of Mycotoxin Mixtures

The dose–effect correlation for the three major type A trichothecenes was determined using the median effect equation based on the mass action law [54,55]:(1)$$\frac{f_{a}}{f_{u}}={\left(\frac{D}{D^{m}}\right)}^{m}$$

In this context, D refers to the concentration of the mycotoxin, fa is the rate of inhibition of cell proliferation, fu is the fraction of cells that remain unaffected (fu = 1 − fa), Dm is the dose that causes a 50% inhibition in cell proliferation (IC_50_), and m is the coefficient that describes the shape of the dose–response curve (with m = 1 for hyperbolic, m > 1 for sigmoidal, and m < 1 for flat sigmoidal curves). This method considers the potency parameter (Dm) for measuring the cytotoxic effects of individual mycotoxins [56].

For systems composed of two or more components, the interactions of various myco-toxins were analyzed using CI methods derived from the median–effect principle (MEP), as described in [54], with CI values determined from the following general equation: (CI)xn=∑j=1nDjDxj.

In this equation, *^n^*(*CI*)*_x_* represents the combination index for n mycotoxins at x% inhibition of cell proliferation; (D)j is the dose of n mycotoxins responsible for x% inhibition in the combination; and (Dx )j is the dose of each n mycotoxin alone that causes x% inhibition of cell proliferation. In general, CI = 1, CI < 1, and CI > 1 indicate additive, synergistic, and antagonistic effects, respectively [19,34].

Moreover, when the mixtures demonstrates a synergistic effect, the dose reduction indices (DRIs) were calculated. The DRI indicates how much the dose of each mycotoxin is reduced in a synergistic combination at a given effect level compared to the individual doses of each mycotoxin. The DRI equation was formulated according to [57]:(2)(CI)xn=∑j=1nDjDxj=∑j=1n1DRIj and DRIj=DjDmj

#### 5.5.3. Analysis of Results

The CalcuSyn software (version 3.0.1; ComboSyn, Inc.; Paramus, NJ, USA) was used to calculate dose–effect curve parameters, CI values, fa-CI plots (showing CI values at different fractions of affected cells), and polygonograms (visual representations of synergism, additive effects, and antagonism for two or three mycotoxin combinations).

### 5.6. Measurement of Intracellular Reactive Oxygen Species

The ROS content in Jurkat cells was measured using an ROS assay kit (Multiskan GO; ThermoFisher; Waltham, MA, USA). Cells (1 × 10^8^/well) were cultured in 96-well plates and treated with T-2, HT-2, DAS, or their dual or triple combinations at IC_50_ concentrations for 24 h. After treatment, the ROS assay kit was applied; subsequently, the plates were incubated at 37 °C for 60 min. Fluorescence was measured at 520 nm using the microplate reader, and ROS levels were expressed as a percentage increase compared to the untreated control (set at 100%) [27].

### 5.7. Evaluation of Apoptosis Assays

After staining the cells with Apopxin™ Green and observing them under a fluorescence microscope, it is crucial in data analysis and interpretation to quantify the extent of apoptotic cell death. This process involves counting fluorescent cells under a fluorescence microscope and interpreting the results in the context of the experiment. The increased Apopxin™ Green positive cells indicated that higher levels of early apoptosis occurred due to phosphatidylserine exposure [27].

### 5.8. Statistical Analysis

The number of viable cells, CI/Fa values, and ROS levels and the apoptotic cell percentage were presented as mean ± standard error of the mean (SEM). Student’s *t*-test was used to compare the means of treated groups at each concentration with the control group. A *p*-value < 0.05 was regarded statistically significant.

## Figures and Tables

**Figure 1 toxins-17-00203-f001:**
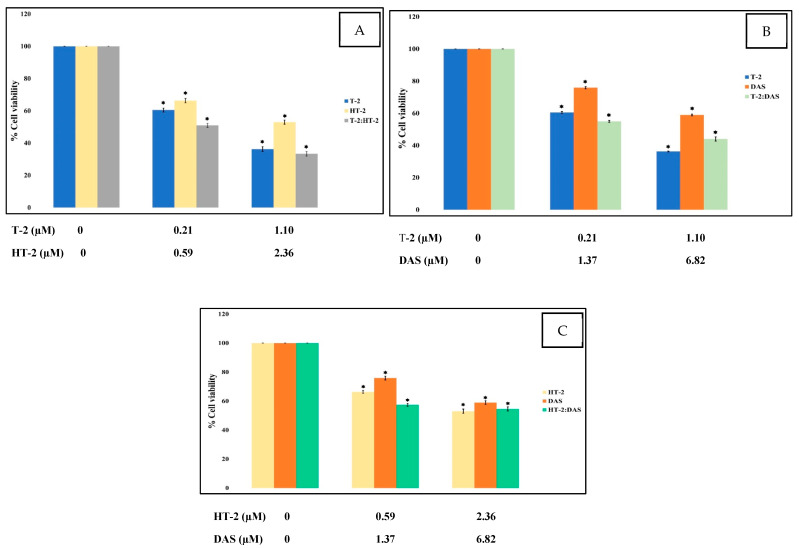
Cytotoxic effects of single mycotoxins T-2, HT-2, or DAS and their dual mixtures on Jurkat T cell survival. (**A**–**C**) Human Jurkat T cells were treated with serial dilutions of each toxin alone (T-2, HT-2, or DAS) or in combinations (T-2 + HT-2, T-2 + DAS, or HT-2 + DAS) for 24 h, and cell viability was assessed based on MTT assays. Data are presented as mean ± SEM values of three experiments. * = *p* < 0.05 compared to controls set to 100%.

**Figure 2 toxins-17-00203-f002:**
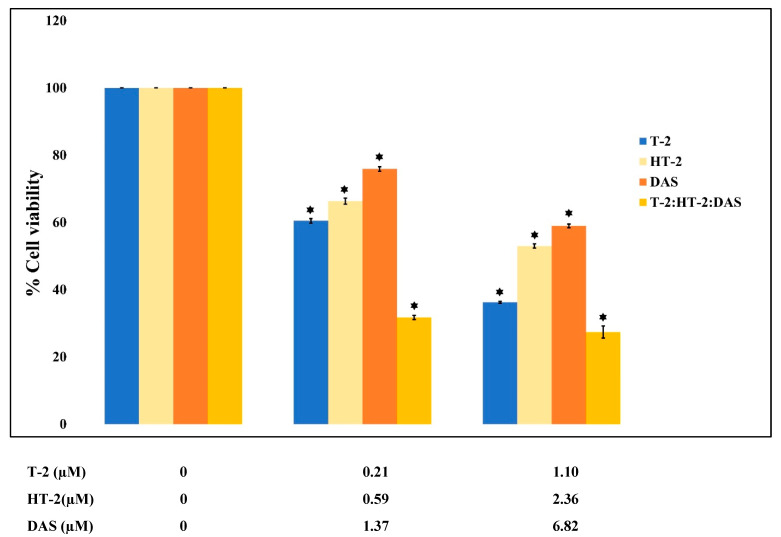
Cytotoxic effects of single mycotoxins T-2, HT-2, or DAS and their triple mixtures on cell viability of human Jurkat T cells. Cells were treated for 24 h, and cell viability was assessed with MTT assays. Data are presented as mean ± SEM values of three experiments. * = *p* < 0.05 compared to controls set to 100%.

**Figure 3 toxins-17-00203-f003:**
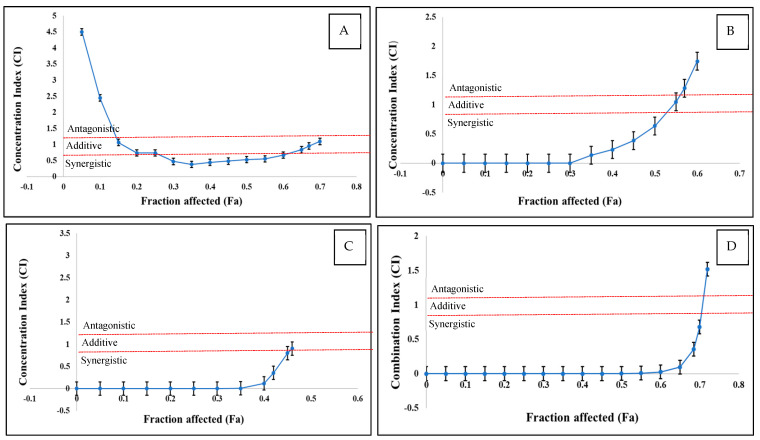
CI versus fractional effect curve (fa), as derived with Chou and Talalay method for human Jurkat T cells exposed to T-2 + HT−2 (**A**), T-2 + DAS (**B**), HT-2 + DAS (**C**), and T-2 + HT-2 + DAS (**D**) over 24 h. Each point represents CI ± SEM of fractional effect. Red dotted line indicates present additive interaction, area under line red dotted line indicates synergism, and area above red dotted line indicates antagonism.

**Figure 4 toxins-17-00203-f004:**
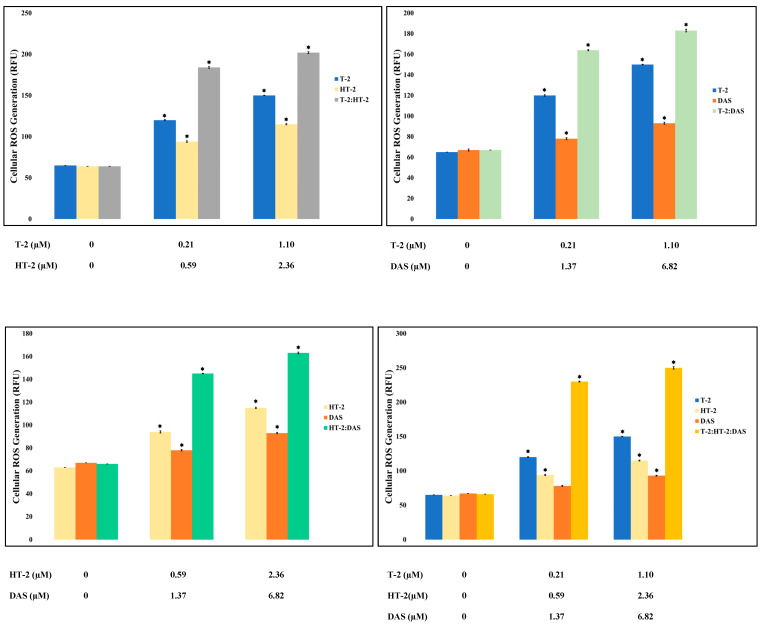
Reactive oxygen species (ROS) levels were measured in Jurkat T lymphocytes following 24 h of exposure to individual mycotoxins (T-2, HT-2, or DAS) as well as to dual or triple toxin mixtures. ROS production increased in dose-dependent manner under all treatment conditions. Results are presented as mean ± SEM (*n* = 3), with * indicating *p* < 0.05 versus control group.

**Figure 5 toxins-17-00203-f005:**
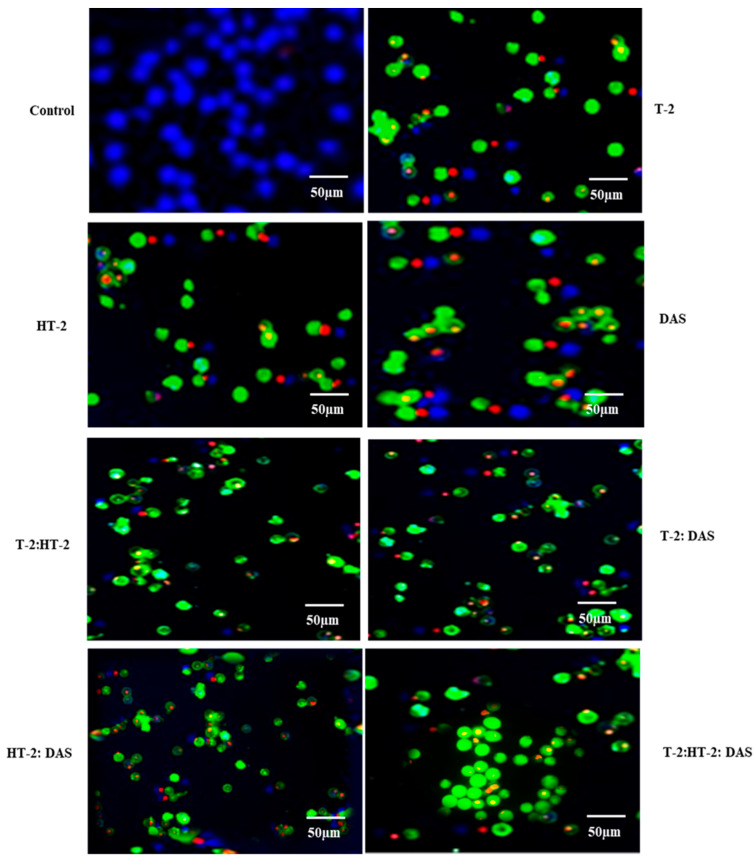
Representative fluorescence microscopy images of Jurkat T lymphocytes displaying viable (blue), apoptotic (green), and necrotic (red) states after treatment with T-2, HT-2, or DAS and their dual or triple combinations for 24 h. Fluorescence images of cells were taken using fluorescence microscope via violet, FITC, and Texas red channels, respectively.

**Figure 6 toxins-17-00203-f006:**
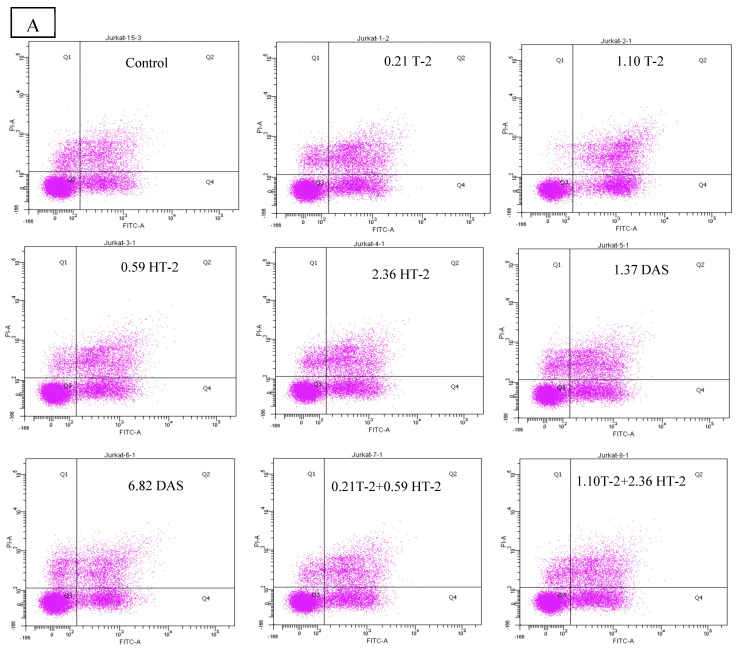
Flow cytometry with Annexin V− FITC and PI staining of human Jurkat T cell clone E6.1 treated with a single mycotoxin treatment (T−2, HT− 2, or DAS, respectively) and their dual or triple combinations for 24 h. Cells were stained and analyzed using flow cytometry to detect Annexin V-FITC− positive and/or PI − positive cells. The cell population was divided into quadrants as follows: viable (lower left), early apoptotic cells (lower right), late apoptotic (upper right), and necrotic cells (upper left), respectively. (**A**) A representative dot plot and (**B**) the percentage of early apoptosis resulting from the indicated treatments. The results are presented as the mean ± SEM values of the data reported in the apoptosis assessment figure. * = *p* < 0.05 compared to the control.

**Table 1 toxins-17-00203-t001:** The parameters of the dose–effect relationship parameters for T-2, HT-2, and DAS and their mixtures in relation to the cell viability of human Jurkat T cells (for individual data, see Appendix A). Dm, m, and r represent median–effect doses, with m indicating the slope of median–effect curves and r indicating the coefficient if linear regression was derived from experimental data according to the mass action law. m > 1, m = 1, and m < 1 indicate hyperbolic, sigmoidal, and negative sigmoidal dose–effect curves, respectively.

Mycotoxin		Dose–Effect Parameter		
	*Dm* (µg/mL)	*Dm* (µM)	*m*	*r*
**T-2** **HT-2** **DAS** **T-2+HT-2** **T-2+DAS** **HT-2+DAS** **T-2+HT-2+DAS**	0.32 1.60 1.46 0.33 0.59 39.56 0.01	0.69 3.77 4.17 0.73 1.38 103.08 0.06	1.32 0.345 1.07 0.48 0.30 0.07 0.10	1.00 1.00 1.00 1.00 1.00 1.00 1.00

**Table 2 toxins-17-00203-t002:** Combination index (CI) and dose reduction index (DRI) values for cytotoxicity caused by individual T-2, HT-2, and DAS treatments and their mixtures in human Jurkat T cells.

Mycotoxin	Joint Ratio	CI and DRI Values (µg/mL)														
		IC_10_	DRI		IC_25_	DRI		IC_50_	DRI		IC_75_	DRI		IC_90_	DRI	
**T-2 +** **HT-2**	**1:2**	1.80800	- -	Ant	0.56323	15.0060 1.94070	Syn	0.47629	3.2574 8.7761	Syn	1.49135	- -	Ant	6.27335	- -	Ant
**T-2 +** **DAS**	**1:5**	0.00278	923.993 589.777	Syn	0.04193	55.1263 42.0375	Syn	0.63780	3.2889 2.9963	Syn	9.77874	- -	Ant	151.115	- -	Ant
**HT-2 +** **DAS**	**1:2.5**	0.000001	0.00005 0.0001	Syn	0.00005	80.16 306.26	Syn	117.643	- -	Ant	9290.400	- -	Ant	9500.5	- -	Ant
**T-2 +** **HT-2 +** **DAS**	**0.1:0.2:0.5**	0.000001	0.005 0.0014 0.00003	Syn	0.00018	0.00018 0.00023 0.00089	Syn	0.00535	238.39 529.86 2172.82	Syn	2.65998	- -	Ant	1548.99	- -	Ant

CI < 1, CI = 1, and CI > 1 indicate synergistic, additive, and antagonistic effects, respectively; DRI > 1 and DRI < 1 indicate supportable and not supportable dose reduction, respectively, while DRI = 1 represents no dose reduction.

## Data Availability

The original contributions presented in this study are included in this article. Further inquiries can be directed to the corresponding author.

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
