# Peer review of "Apoptotic Effect of Combinations of T-2, HT-2, and Diacetoxyscirpenol on Human Jurkat T Cells"

_toxins, 2025, doi:10.3390/toxins17040203_

Round 1

Reviewer 1 Report

Comments and Suggestions for Authors

The manuscript "toxins-3535665-Combined cytotoxicity of T-2, HT-2, and diacetoxyscirpenol 2mixture on human Jurkat T cells" submitted to journal Toxins brings description about cytotoxic effect of three Trichothecene type A mycotoxins (alone and as mixtures) on Jurkat cells. The manuscript is well written and easy to follow.

General comments:

The manuscript is well written and obtained results clearly support the conclusions drawn, but a few things should be noted:

  1. Please uniform the terminology for combinations: choose either “dual and triple” or “binary and ternary”.
  2. Please include the chemical structures of the three tested mycotoxins in the Materials & Methods section.
  3. Mitochondrial dysfunction is not directly assessed (although some inferences may be drawn from the MTT assay, it is a measure of general toxicity rather than dysfunction). Proper assessment of mitochondrial dysfunction requires more specific tests. Please remove the statement in the conclusion suggesting that dysfunction was evaluated.
  4. Please remove the extra space in line 67.
  5. Please use “h” for “hour” instead of “hr”.
  6. Please add space in “Figure1” in line 86.
  7. Please describe and justify in the text (either in the Materials & Methods section or elsewhere) the rationale for selecting specific concentrations of each mycotoxin for the cell viability assay (section 5.2).
  8. Please clarify why all control (0) values are set to 100%. Typically, control values vary between experiments and duplicates. Were these values measured, or were they arbitrarily set at 100%? Additionally, why are there three different colors for control (0) bars? Since they represent the same control, there should be a single bar.
  9. Please provide additional details regarding the concentrations presented in the results section on page 5. It would be beneficial to include a table comparing concentrations in µM and their corresponding values in µg/mL for all individual mycotoxins and combinations.
  10. On page 5, what do the µg/mL concentrations represent? Do they indicate the sum of all mycotoxins, or do they refer to individual components? Please clarify.
  11. Were the parameters in Table 1 measured or calculated? It would be helpful to include a supplementary figure (or an appendix) illustrating how the data were extracted. Additionally, please provide the appropriate units for each parameter where possible.
  12. Please specify the reference or guideline used to define the CI threshold values distinguishing between antagonistic, additive, and synergistic effects.
  13. The results presented focus solely on the "early" phase apoptosis. What about the "late" phase?
  14. In combined treatments, is the vehicle concentration slightly higher than 0.4%, given that all mycotoxins were prepared from 1 mM DMSO stock solutions? Please verify.
  15. Please use “changed” instead “refreshed” in line 515.
  16. Please correct typo “105” to “10^5” (put it in superscript through Materials & Methods section).
  17. Please cite references for the methods used, as they were not developed in this study.
  18. In line 554, is the stated concentration of 100 000 cells/mL in a 6-well plate correct? This seems disproportionately high compared to the number of cells used in a 24-well plate. Furthermore, the manuscript inconsistently refers to either the number of seeded cells or their concentration in different sections, please uniform terminology.
  19. Please format “IC50” correctly by placing “50” in subscript throughout the text.

Author Response

Dear Reviewers,

We appreciate the time and effort the reviewer has put into evaluating our manuscript. We have carefully considered all your suggestions and made the necessary revisions to improve the manuscript accordingly. Please find our detailed responses to each comment below.

Reviewer: 1#

General comments to the Author

  1. Please uniform the terminology for combinations: choose either “dual and triple” or “binary and ternary”.

Response: The authors would like to thank reviewer for your suggestion. The authors standardized the terminology for dual and triple combinations, ensuring consistency throughout the manuscript.

--------------------------------------------------------------------------------------------------------

  1. Please include the chemical structures of the three tested mycotoxins in the Materials & Methods section.

Response:  The authors included the chemical structures of the three tested mycotoxins in Materials and Methods accordingly. (P15, L442-443)

--------------------------------------------------------------------------------------------------------
3. Mitochondrial dysfunction is not directly assessed (although some inferences may be drawn from the MTT assay, it is a measure of general toxicity rather than dysfunction). Proper assessment of mitochondrial dysfunction requires more specific tests. Please remove the statement in the conclusion suggesting that dysfunction was evaluated.

Response: Accordingly, the authors removed the statement in conclusion suggesting that dysfunction in agreement with the reviewer’s suggestion.

--------------------------------------------------------------------------------------------------------
4. Please remove the extra space in line 67.

Response: The extra space was removed in the revised manuscript.

--------------------------------------------------------------------------------------------------------

5. Please use “h” for “hour” instead of “hr”.

Response: We appreciate it very much, the word of “h” was replaced throughout the revised manuscript.

--------------------------------------------------------------------------------------------------------

6. Please add space in “Figure1” in line 86.

Response: The space was added in Figure 1. (P3, L96)

--------------------------------------------------------------------------------------------------------

7. Please describe and justify in the text (either in the Materials & Methods section or elsewhere) the rationale for selecting specific concentrations of each mycotoxin for the cell viability assay (section 5.2).

Response: The various concentrations treated with T-2 toxin, HT-2 toxin and DAS based on IC50 of the MTT result from our previous publication.

    - Wattanasuntorn, P.; Phuektes, P.; Poapolathep, S.; Mimapan, S.; Tattiyapong, M.; Fink-

      Gremmels, J.; Oswald, I.P.; Poapolathep, A. Individual cytotoxicity of three major type    

      A trichothecene, T-2, HT-2, and diacetoxyscirpenol in human Jurkat T cells. Toxicon.

  1. 28(243), 107718.

--------------------------------------------------------------------------------------------------------

8. Please clarify why all control (0) values are set to 100%. Typically, control values vary between experiments and duplicates. Were these values measured, or were they arbitrarily set at 100%? Additionally, why are there three different colors for control (0) bars? Since they represent the same control, there should be a single bar.

Response: In all studies, the same number of cells were seeded in all the wells in the plate, no matter control or treated. Then the average viability in the control wells determined by MTT assay was used as the reference 100% viability, compared to treated wells. The different colors for the control bars represent separate control groups for each type of mycotoxin, rather than a single control group for all results.

--------------------------------------------------------------------------------------------------------

9. Please provide additional details regarding the concentrations presented in the results section on page 5. It would be beneficial to include a table comparing concentrations in µM and their corresponding values in µg/mL for all individual mycotoxins and combinations.

Response:  In line with the reviewer’s suggestion, we included the table comparing concentrations in µM and their corresponding values in µg/mL for all individual mycotoxins and combinations in Table 1. In the context of Dose-Effect relationships, the parameters Dm, m, and r are part of the Hill equation and are often used to describe the relationship between the dose of a substance (like a toxin, drug, or chemical) and the effect it produces. These parameters are especially important when modeling and analyzing biological responses to various doses of substances.

  • Dm represents the dose (or concentration) at which the biological effect is 50% of the maximum effect
  • m is the Hill coefficient, also called the slope factor or Hill slope.
  • r is a linear correlation coefficient that manifests the conformity of the experimental data to the median-effect plot of the mass-action law. r = 1 indicates perfect conformity.

-----------------------------------------------------------------------------------------------

10. On page 5, what do the µg/mL concentrations represent? Do they indicate the sum of all mycotoxins, or do they refer to individual components? Please clarify.

Response: Accordingly, the concentrations in µg/mL (shown in parentheses) represent dose-effect relationships within the combined toxicity profiles of the studied toxins, rather than the effects of individual mycotoxins. The Dm values for mixtures represent the cumulative concentrations (in µg/mL) of all the toxins in the mixture that exert 50% of the maximum biological effect.

--------------------------------------------------------------------------------------------------------

11. Were the parameters in Table 1 measured or calculated? It would be helpful to include a supplementary figure (or an appendix) illustrating how the data were extracted. Additionally, please provide the appropriate units for each parameter where possible.

Response: The parameters presented in Table 1. were calculated using dose-effect relationship graphs, which are included in the supplementary S1 in agreement with the reviewer’s comments.

--------------------------------------------------------------------------------------------------------

12. Please specify the reference or guideline used to define the CI threshold values distinguishing between antagonistic, additive, and synergistic effects.

Response: Accordingly, the Combination Index (CI) threshold values to determine drug interactions—whether they are synergistic, additive, or antagonistic—are typically based on the method developed by Ting-Chao Chou and Paul Talalay in the Median-Effect Principle and Combination Index-Isobologram Equation (Chou-Talalay Method; Alassane-Kpembi et al., 2017).

--------------------------------------------------------------------------------------------------------

13.The results presented focus solely on the "early" phase apoptosis. What about the "late" phase?

Response: The focus of this study was on apoptosis. The early stage analysis is the only reliable frame to discriminate apoptosis from necrosis using flow cytometry. In the early stage of apoptosis, phosphatidylserine, the biomarker of apoptosis, is translocated from the inside of the cell membrane to the outside, while the membrane still keeps its integrity. Phosphatidylserine can then be stained by annexin V and quantified unequivocally. Later in the cell death process, once the membrane integrity is lost, it is no more possible to discriminate the translocated phosphatidylserine from the inner membrane phosphatidylserine, and no reliable difference can be made between apoptotic and necrotic cells. The late phases are therefore not relevant for a reliable analysis of apoptosis.

--------------------------------------------------------------------------------------------------------

14. In combined treatments, is the vehicle concentration slightly higher than 0.4%, given that all mycotoxins were prepared from 1 mM DMSO stock solutions? Please verify.

Response: Thank you very much for your comments. In combined treatments, the vehicle (DMSO) concentration is slightly higher than 0.4% unless adjustments are made to keep it constant. If each mycotoxin is added from a separate DMSO stock, then the total DMSO amount will be slightly higher than in single-toxin treatments. To verify precisely, you should calculate the final DMSO percentage based on the total volume of all mycotoxin solutions added.

--------------------------------------------------------------------------------------------------------
15. Please use “changed” instead “refreshed” in line 515.

Response: The word of “changed” was replaced in the revised manuscript. (P15, L457)

--------------------------------------------------------------------------------------------------------

16. Please correct typo “105” to “10^5” (put it in superscript through Materials & Methods section).

Response: Thank you very much. The number was corrected through Materials and Methods section of the revised manuscript.

--------------------------------------------------------------------------------------------------------
17. Please cite references for the methods used, as they were not developed in this study.

Response: The methods used in this study, including the apoptosis detection assay (Annexin V/PI staining), cytotoxicity assessment (MTT assay), and dose-effect analysis (median-effect principle), were based on previously established protocols. Specifically, the apoptosis detection method follows the approach described by Wattanasuntorn et al. (2024), the dose-effect relationship was analyzed using the median-effect method developed by Chou & Talalay (1984). The references were cited for the methods used in the revised manuscript.

--------------------------------------------------------------------------------------------------------

18. In line 554, is the stated concentration of 100 000 cells/mL in a 6-well plate correct? This seems disproportionately high compared to the number of cells used in a 24-well plate. Furthermore, the manuscript inconsistently refers to either the number of seeded cells or their concentration in different sections, please uniform terminology.

Response: In this study, a concentration of 1 × 10⁵ cells/mL in a 6-well plate was used, following the recommendations of the commercial kit protocol. Given a total volume of 2 mL per well, this results in 2 × 10⁵ cells per well. This concentration was chosen based on the kit's validated methodology to ensure optimal cell viability and reliable assay performance.

--------------------------------------------------------------------------------------------------------

19. Please format “IC50” correctly by placing “50” in subscript throughout the text

Response: We placed 50 in subscript throughout the revised manuscript.

--------------------------------------------------------------------------------------------------------

Reviewer 2 Report

Comments and Suggestions for Authors

Individual and combination toxicity of trichothecene type A mycotoxins (T-2, HT-2, and DAS) have been investigated in this work. The results showed synergistic effects at low concentrations and an apparent antagonism at very high concentrations for all combinations of the three trichothecenes. The results are very interesting but lack of antagonism investigations.

  1. How about the contamination level of the three mycotoxins in foods and feeds? Please add recent reports in the manuscript.
  2. How does the author decide the levels of the three mycotoxins investigated in the work?
  3. What do you mean t low concentrationsand very high concentrations in the abstract?
  4. It is mentioned that “The additional cytometric analyses confirmed the
  5. synergistic effects”. How about the antagonism at very high concentrations for all combinations of the three trichothecenes?
  6. Figure 1. Please combine the three pic in one.
  7. Please explain why Jurkat T cells were applied for cytotoxic effects investigation?
  8. From figure 2, it seems that cytotoxic effects induced the combination of the three mycotoxins are higher than each at both tested concentrations? How does the authors get the conclusion “antagonism at very high concentrations”?

Other comments:

  1. Please cite more recent references. Most are over ten years ago.

Author Response

Dear Reviewers,

We appreciate the time and effort the reviewer has put into evaluating our manuscript. We have carefully considered all your suggestions and made the necessary revisions to improve the manuscript accordingly. Please find our detailed responses to each comment below.

Reviewer  2#

General comments to the Author

1. How about the contamination level of the three mycotoxins in foods and feeds? Please add recent reports in the manuscript.

Response: ​Recent studies indicate that the contamination levels of T-2 and HT-2 toxins in foods and feeds vary significantly across different regions and commodities. In Northern Europe, particularly in the United Kingdom, oats have exhibited high concentrations of these toxins, with mean levels reaching up to 570 µg/kg and maximum levels as high as 9,990 µg/kg. Similarly, in Scotland, a 2019 survey found T-2 and HT-2 toxins in 91% of oat samples, with concentrations ranging from non-detectable to 3,474 µg/kg. In contrast, data from Thailand between 2015 and 2020 show that T-2 toxin contamination in animal feeds was predominantly low, with 94.7% of samples containing less than 25 µg/kg and no samples exceeding 250 µg/kg. Information on diacetoxyscirpenol contamination is less prevalent, but it is generally detected at lower frequencies and concentrations compared to T-2 and HT-2 toxins [6]. (P1, L40; P2, L41-51)

--------------------------------------------------------------------------------------------------------
2. How does the author decide the levels of the three mycotoxins investigated in the work?

Response: The authors would like to thank reviewer for your question. The levels of the three mycotoxins (T-2, HT-2, and DAS) were determined based on IC50 of previous studies, regulatory guidelines, and their known toxicological effects in human cell lines. The concentrations were chosen to reflect physiologically relevant exposure levels, while also allowing for dose-dependent comparisons of cytotoxic and apoptotic effects in Jurkat T cells. Additionally, these levels were selected to ensure compatibility with the experimental design, enabling the evaluation of both individual and combined effects of the mycotoxins.

--------------------------------------------------------------------------------------------------------

3. What do you meant low concentrations and very high concentrations in the abstract?

Response: In this study, low concentration indicated that these typically refer to doses that are below or near the no observed effect level or sub – toxic levels (IC10 to < IC50) where minimal effects on cell viability responses are observed. In addition, high concentrations (IC50 to IC90) are cytotoxic doses, usually close to or exceeding the IC₅₀ (half-maximal inhibitory concentration)

--------------------------------------------------------------------------------------------------------

4. It is mentioned that “The additional cytometric analyses confirmed the synergistic effects”. How about the antagonism at very high concentrations for all combinations of the three trichothecenes?

Response: We appreciate the reviewer's comment. In our study, we focused on synergistic effects because they have significant toxicological relevance, especially in risk assessment and regulatory settings. Synergism suggests that even low concentrations of mycotoxin mixtures could be more harmful than previously estimated. Additionally, synergistic interactions provide valuable mechanistic insights into cellular toxicity. While antagonism at very high concentrations was observed, it may result from saturation effects or cellular stress responses, making it more challenging to interpret mechanistically. Further studies are needed to fully characterize these high-dose interactions.

--------------------------------------------------------------------------------------------------------

5.Figure 1. Please combine the three pic in one.

Response: The authors would like to thank the reviewer for pointing out and recommendation. The Figure 1 was amended in the revised manuscript as the reviewer’s recommendation.

------------------------------------------------------------------------------------------------------------------
6. Please explain why Jurkat T cells were applied for cytotoxic effects investigation?

Response: Jurkat T cells were applied for investigating the cytotoxic effects of T-2, HT-2, and diacetoxyscirpenol (DAS) because they serve as a well-established model for studying immune-related toxicity. These mycotoxins, produced by Fusarium fungi, are known to suppress immune function by inhibiting protein synthesis and inducing apoptosis in lymphocytes. Jurkat cells, derived from human T lymphocytes, provide an ideal system to evaluate the immunosuppressive effects of these toxins at the cellular level. Their sensitivity to oxidative stress, mitochondrial dysfunction, and DNA damage makes them a suitable model to examine the molecular mechanisms underlying mycotoxin-induced cytotoxicity. Additionally, as T cells play a crucial role in immune responses, studying their viability and function upon toxin exposure helps in understanding how these contaminants may contribute to immunosuppression in humans and animals consuming contaminated food or feed. 

    Moreover, Jurkat cells are widely used in toxicological research due to their ease of culture and ability to rapidly proliferate in suspension, making them ideal for high-throughput screening of toxic compounds. Their well-characterized signaling pathways, including those involved in apoptosis (such as caspase activation and mitochondrial dysfunction), allow researchers to dissect the mechanisms by which T-2, HT-2, and DAS exert cytotoxic effects. These toxins are known to induce oxidative stress and disrupt intracellular signaling, leading to cell cycle arrest and apoptosis, which can be efficiently studied in Jurkat cells. Additionally, their use enables comparison with other immune and non-immune cell lines, providing insight into tissue-specific toxic effects.

--------------------------------------------------------------------------------------------------------
7. From figure 2, it seems that cytotoxic effects induced the combination of the three mycotoxins are higher than each at both tested concentrations? How does the authors get the conclusion “antagonism at very high concentrations”?

Response: We appreciate the reviewer’s observation. The combination of the three mycotoxins indeed induced higher cytotoxic effects than each mycotoxin alone at both tested concentrations. However, at very high concentrations, we observed a reduction in the expected combined toxicity, which we interpret as antagonism. This could be due to cytotoxic saturation, where further increases in toxin concentration do not result in a proportional increase in cell death. Additionally, high concentrations may trigger cellular stress responses or detoxification mechanisms that reduce the overall toxic effect. Further investigations are needed to better understand these high-dose interactions and their underlying mechanisms. As shown by the Fa-CI plots in Fig.3, CI was >1, which indicates antagonism.
--------------------------------------------------------------------------------------------------------
8. Please cite more recent references. Most are over ten years ago.

Response: Authors would like to thank reviewer for your suggestion. Accordingly, we added the references as the reviewer’s suggestion.

--------------------------------------------------------------------------------------------------------

Round 2

Reviewer 1 Report

Comments and Suggestions for Authors

I don't have additional comments.

Reviewer 2 Report

Comments and Suggestions for Authors

Thanks for the response. All the comments have been revised and I would recommend accept in the current version